# Development of Nanocomposite Microspheres for Nasal Administration of Deferiprone in Neurodegenerative Disorders

**DOI:** 10.3390/jfb15110329

**Published:** 2024-11-05

**Authors:** Radka Boyuklieva, Plamen Katsarov, Plamen Zagorchev, Silviya Abarova, Asya Hristozova, Bissera Pilicheva

**Affiliations:** 1Department of Pharmaceutical Sciences, Faculty of Pharmacy, Medical University of Plovdiv, 4002 Plovdiv, Bulgaria; radka.boyuklieva@mu-plovdiv.bg (R.B.); plamen.katsarov@mu-plovdiv.bg (P.K.); asya.hristozova@mu-plovdiv.bg (A.H.); 2Research Institute, Medical University of Plovdiv, 4002 Plovdiv, Bulgaria; plamen.zagorchev@mu-plovdiv.bg; 3Department of Medical Physics and Biophysics, Faculty of Pharmacy, Medical University of Plovdiv, 4002 Plovdiv, Bulgaria; 4Department of Medical Physics and Biophysics, Faculty of Medicine, Medical University of Sofia, 1000 Sofia, Bulgaria; silviya.m.abarova@gmail.com; 5Department of Analytical Chemistry and Computational Chemistry, Faculty of Chemistry, University of Plovdiv “Paisii Hilendarski”, 4000 Plovdiv, Bulgaria

**Keywords:** deferiprone, nanocomposite microspheres, nasal administration, nose-to-brain delivery, neurodegenerative disorders

## Abstract

Elevated brain iron levels are characteristic of many neurodegenerative diseases. As an iron chelator with short biological half-life, deferiprone leads to agranulocytosis and neutropenia with a prolonged therapeutic course. Its inclusion in sustained-release dosage forms may reduce the frequency of administration. On the other hand, when administered by an alternative route of administration, such as the nasal route, systemic exposure to deferiprone will be reduced, thereby reducing the occurrence of adverse effects. Direct nose-to-brain delivery has been raised as a non-invasive strategy to deliver drugs to the brain, bypassing the blood–brain barrier. The aim of the study was to develop and characterize nanocomposite microspheres suitable for intranasal administration by combining nano- and microparticle-based approaches. Nanoparticles with an average particle size of 213 ± 56 nm based on the biodegradable polymer poly-ε-caprolactone were developed using the solvent evaporation method. To ensure the deposition of the particles in the nasal cavity and avoid exhalation or deposition into the small airways, the nanoparticles were incorporated into composite structures of sodium alginate obtained by spray drying. Deferiprone demonstrated sustained release from the nanocomposite microspheres and high iron-chelating activity.

## 1. Introduction

Diseases of the central nervous system (CNS) are still a major challenge to the medical community. Most of the therapeutic agents administered orally or parenterally cannot reach the brain tissue due to the presence of the blood–brain and blood–cerebrospinal barriers. The latter, in combination with the presystemic metabolism, reduces the efficacy of the therapy, requires high doses of medicinal substances, and often causes adverse side effects [1,2,3].

Metals such as iron (Fe) and copper (Cu) are known to catalyze redox processes in the body. Longer lifespans lead to the accumulation of iron in various parts of the brain, although this is not always associated with toxicity. Iron plays a key role in the physiological functions of the brain, being involved in mitochondrial respiration and myelin and neurotransmitter synthesis. The distribution of iron in the brain is heterogeneous, with the highest concentrations found in the substantia nigra pars compacta and the basal ganglia, where it reaches levels comparable to those in the liver [4]. Various inflammatory processes increase the permeability of the blood–brain barrier and lead to Fe accumulation in the CNS. It is not fully understood why iron accumulation in specific brain regions becomes highly toxic in neurodegenerative diseases. Recent studies of activated microglia have shown that these cells exhibit a propensity to accumulate Fe, which can subsequently lead to cell death [5,6,7]. This type of cell death mediated by Fe-dependent lipid peroxidation is known as ferroptosis [8]. As cells age, their ability to maintain redox balance declines, leading to the accumulation of free reactive oxygen species (ROS), mitochondrial dysfunction, and neuronal damage.

Deferiprone (DFP) is an iron chelator and is indicated for the treatment of iron overaccumulation in patients with thalassemia major [9]. It belongs to the family of alpha-keto hydroxy pyridines, which are a relatively new class of chelating agents (Figure 1A). DFP binds iron in a molar ratio of 3:1 (Figure 1B). It is one of the few chelators that can cross cellular and subcellular membranes and enter cell organelles, where it binds to accumulated iron reserves and transports the complex-bound iron to plasma transferrin [10]. DFP can bind and eliminate accumulated iron from various parts of the body, including the liver, brain, and heart. DFP exhibits selectivity for iron and almost no affinity for other metals such as zinc, aluminum, and copper. After oral administration, DFP is rapidly and completely absorbed from the upper compartments of the gastrointestinal tract. Peak plasma concentrations of DFP taken in the fasting state are reached approximately 1 h after oral administration. Its elimination occurs mainly by metabolism in the liver, and the major metabolite is a 3-O-glucuronide conjugate that is excreted by the kidney. This metabolite has no iron-binding capacity due to the inactivation of the 3-hydroxy group of DFP. It has a biological half-life of 2–3 h, necessitating the maintenance of effective plasma drug concentrations by administering individual doses at short intervals several times daily [11]. This can be a cause of non-adherence, especially in elderly patients, and compromise treatment success. The development of sustained-release dosage forms of DFP may reduce dosing frequency and improve patients’ compliance.

In view of the drawbacks of the oral route, researchers turned to alternative routes of administration. Intranasal administration of drugs eliminates the problems associated with their low absorption, reduced stability, or rapid biotransformation. The olfactory region, located in the hard-to-reach upper part of the nasal cavity, allows substances to bypass the blood–brain barrier and enter the brain directly. This occurs due to the olfactory neurons being in contact with the nasal cavities and the central nervous system simultaneously [12]. Direct nasal administration of some drugs without the use of carriers can provoke serious adverse clinical manifestations. To reach the brain intact, therapeutic agents must be incorporated into appropriate drug delivery systems. Nanosized carriers can carry drugs from the nasal cavity to the brain while protecting them from enzymatic and/or chemical degradation and improving their passage across membrane barriers [13]. Despite the listed advantages, the nanosized structures, due to their small size, follow the respiratory tract and exit the nasal cavity after nasal administration, while microparticles of about 10 µm size after inhalation can deposit in the olfactory region of the nasal cavity [14]. For smaller particles, the site of deposition depends on the velocity at which the particles are inhaled and the turbulence in the airflow [15]. There is some evidence that microparticles improve the delivery of active molecules to the CNS by prolonging the contact time with the nasal mucosa, but due to their larger size, they cannot reach the brain and release the active molecules there [16,17]. Optimizing the size of drug-loaded particles will help their preferential deposition in the olfactory region, and the contact time between the particles and the mucosa will be crucial for the release and absorption of drugs [18,19].

Researchers are expanding the scope of synthesis from basic structures, such as micro- and nanoparticles, to more complex carriers. Nanocomposite microparticles are a new class of drug delivery systems that consist of nanosized particles dispersed into a polymer matrix. By designing the composite structure, carriers with specific physicochemical and mechanical characteristics can be obtained [20]. These structures are a combination of the best properties of their components, exhibiting additional interesting characteristics that the individual ingredients often do not possess.

The aim of the present study is to develop nanocomposite microspheres with optimal bioadhesive and biopharmaceutical properties for the nasal administration of DFP in the therapy of neurodegenerative diseases.

## 2. Materials and Methods

### 2.1. Materials

Deferiprone (DFP, Mw 139.15 g/mol), poly-ε-caprolactone (PCL, Mw 14,000, 80,000 g/mol), sodium alginate (from brown algae, medium viscosity ≥ 2000 cP, 2% (25 °C)), polysorbate 20 (Mw 1220 g/mol), Span 85 (Mw 957.49 g/mol), and human serum albumin (≥96%) were purchased from Sigma-Aldrich (St. Louis, MO, USA). All other reagents and solvents were of analytical grade and used as provided.

### 2.2. Methods

#### 2.2.1. Nanoparticles (NP) Preparation

For the nanoencapsulation of water-soluble substances (such as DFP) by a solvent evaporation technique, the preparation of a W/O emulsion is appropriate. Due to the difficulty in isolating the particles from the oil phase of the emulsion and with a view to their subsequent incorporation into composite structures in the form of aqueous dispersions, the double emulsion technique and the preparation of a W/O/W emulsion type were chosen. The preparation of PCL nanoparticles by double emulsion technique with subsequent evaporation of the solvent was carried out in two steps (Figure 2). In the first step, for the formation of the primary emulsion (W1/O), the internal aqueous phase (W1) without active substance (placebo) or with DFP was emulsified in the oil phase containing the polymer (PCL) dissolved in dichloromethane (DCM) and the emulsifying agent (Span 85) under high-speed homogenization at 17,500 rpm (Miccra MiniBatch D-9, MICCRA GmbH, Heitersheim, Germany) for 3 min. In the second step, the primary emulsion (W1/O) was dispersed into 50 mL of the external aqueous phase (W2) containing purified water and polysorbate 20 as an emulsifying agent. Emulsification was performed under high-speed homogenization at 22,500 rpm for 5 min to form the double emulsion (W1/O/W2). Subsequently, the formed nanoemulsion was stirred with an HS-100D mechanical stirrer (Witeg Labortechnik GmbH, Wertheim, Germany) at a speed of 800 rpm until complete evaporation of dichloromethane, resulting in solidification of the PCL polymer droplets. The obtained NPs were separated by centrifugation in Amicon^®^ ultracentrifuge tubes (Merck KGaA, Darmstadt, Germany) at 3800× *g* for 20 min at (20 ± 1) °C (Figure 2), washed with petroleum ether to remove residual water, and dried at room temperature for 24 h. The samples were stored in tightly closed containers in a dark place at 20 ± 2 °C for further use.

#### 2.2.2. Preparation of Nanocomposite Microspheres

Alginate microspheres loaded with the formulated PCL nanoparticles were prepared by spray drying using Büchi Mini Spray Dryer B-290 (Büchi Labortechnik AG, Flawil, Switzerland). Briefly, sodium alginate was dissolved in purified water (0.5% or 1% *w*/*v*), and the drug-loaded nanoparticles at an amount corresponding to 200 mg drug content were dispersed into the polymer solution (100 mL). The dispersions were continuously stirred on a magnetic stirrer at 300 rpm and were spray–dried through a 0.7 mm nozzle using pressurized nitrogen (5 bar) and aspiration of 35 m^3^/h. To achieve a high production yield and mean particle diameter above 5 µm (suitable for nasal administration) [21], the process parameters were varied as follows: drying temperature from 120 °C to 160 °C, feed rate 5 ÷ 7.5 mL/min, and gas flow rate from 400 to 600 L/h. Placebo nanocomposite microparticles without drug substance were also prepared using the described methodology.

### 2.3. Characterization Techniques

#### 2.3.1. Particle Size Analysis, Size Distribution, and Zeta Potential

The size, size distribution, and ζ-potential of the obtained nanoparticles were determined by a Nanotrac particle size analyzer (Microtrac, York, PA, USA) provided with a 3 mW He/Ne laser at 780 nm wavelength. Noninvasive backscattering technology allows for analyzing the size in the range of 0.8 nm to 6.5 µm. The samples from the nanoparticle suspension were directly measured. All the measurements were performed at 25.0 °C at 20 s intervals and were repeated three times.

The size of the nanocomposite microparticles and their size distribution were determined by laser diffraction with an LS 13 320 particle size analyzer (Beckman Coulter, Brea, CA, USA) equipped with a Tornado Dry Powder System. One hundred milligrams of particles were used for each measurement (*n* = 3).

#### 2.3.2. Scanning Electron Microscopy (SEM)

The formulated nano- and micro-sized particles were analyzed in terms of shape and surface morphology using a scanning electron microscope (Prisma E SEM, Thermo Scientific, Waltham, MA, USA). The dry samples were coated with gold using a sputter coater (Q150T ES Plus, Quorum technologies, West Sussex, UK) and then micrographic images at appropriate magnification were recorded at 15 kV acceleration voltage.

#### 2.3.3. Transmission Electron Microscopy (TEM)

The obtained nanoparticles were also observed using a transmission electron microscope Talos F200C G2 (Talos 1.15.3, Thermo Fisher Scientific, Waltham, MA, USA). The analyzed sample (a suspension of the nanoparticles) was dried for 24 h on a formvar-coated copper grid (200 mesh). The images were taken at 200 kV and recorded with Velox Imaging Software (Velox 2.15.0.45, Waltham, MA, USA).

#### 2.3.4. Fourier-Transform Infrared Spectroscopy (FTIR)

FTIR analysis using a Nicolet iS 10 FTIR spectrometer (Thermo Fisher Scientific, Pittsburgh, PA, USA) was performed in order to evaluate drug–polymer interactions. An ATR accessory (diamond attenuated total reflection) was used to record the spectra of the samples within the range of 600–4000 cm^−1^. OMNIC^®^ software package (Version 7.3, Thermo Electron Corporation, Madison, WI, USA) was utilized for image editing.

#### 2.3.5. Differential Scanning Calorimetry (DSC)

Thermal analysis of the particles was performed using a DSC 204F1 Phoenix (Netzsch Gerätebau GmbH, Selb, Germany). An indium standard (Tm = 156.6 °C, ΔHm = 28.5 J/g) was used to calibrate the temperature and heat flux. The measurements were performed in an argon atmosphere at a heating rate of 10 °C/min.

#### 2.3.6. Production Yield

The production yield of nano- and microparticles was calculated based on the weight of particles obtained (W1) and the total weight of the polymer/polymers (PCL/PCL and sodium alginate) and the drug (W2) by the following equation:$$Yield\;\left(\%\right)=\frac{W1}{W2}\times 100$$

#### 2.3.7. Estimation of Drug Loading (DL) and Entrapment Efficiency (EE)

The amount of DFP included in the nanoparticles and nanocomposite microspheres was determined by high-performance liquid chromatography (HPLC). To separate drug-loaded nanoparticles from the free DFP, the nanoparticle suspension was centrifuged (Sigma Centrifuge 3-18 KS, Osterode am Harz, Germany) in Amicon^®^Ultra-15 tubes (100 K MWCO) (Merck KGaA, Darmstadt, Germany) for 20 min at 3800× *g* at 20 ± 1 °C. Dichloromethane, methanol, and purified water were added to the nanoparticles from each batch. The samples were placed in a sonication bath (Sonorex Bandelin electronic, Berlin, Germany) to extract DFP. After filtration (PTFE, 0.45 μm), the amount of DFP was determined by HPLC (UltiMate 3000, Thermo Scientific, Waltham, MA, USA) using an Inertsil^®^ ODS-3HPLC column (150 × 4.6 mm, 5 µm, GL Sciences, Tokyo, Japan) under the following conditions: wavelength 280 nm, injection volume 50 μL, flow rate 1 mL/min, column oven temperature 22 °C, and mobile phase PBS pH 2.5 (o-H_3_PO_4_) and acetonitrile in a ratio 67:33 (*v*:*v*). DFP retention time was 4.86 min; assay duration was 15 min.

To evaluate the DEE of composite structures, dichloromethane, methanol, and water were added to the microparticles. They were placed in a sonication bath to extract DFP. After filtration (PTFE, 0.45 μm), the concentration of DFP was determined according to the described HPLC method.

The drug loading (DL) of the particles was calculated according to the equation:$$DL\left(\%\right)=\frac{Amount\;of\;drug\;in\;the\;formulation\;(mg)}{Total\;amount\;of\;particles\;(mg)}\times 100$$

The drug entrapment efficiency (DEE) was calculated according to the following equation:$$DEE\left(\%\right)=\frac{Actual\;drug\;content\;(mg)}{Theoretical\;drug\;content\;(mg)}\times 100$$

#### 2.3.8. In Vitro Drug Release Studies and Release Kinetics

The drug release behavior was studied in vitro using the dialysis bag diffusion technique. Dialysis tubing cellulose membrane (MWCO 12 kDa, Sigma-Aldrich Chemie GmbH, Taufkirchen, Germany) was used. One day before the study, the dialysis membrane was cut into pieces 6 × 2.5 cm^2^ and hydrated in purified water. In the dialysis bag, 1 mL dispersion of nanoparticles (a certain amount, equivalent to 1.15 mg DFP) in phosphate-buffered saline (PBS, pH 7.4) was placed and the bag was closed with plastic clamps. The dialysis bag was placed in a glass beaker containing an 18 mL acceptor medium (PBS buffer, pH 7.4) and kept on an electromagnetic stirrer at 100 rpm and 37 °C. At predetermined time points, 1 mL from the release medium was withdrawn and replaced with an equal volume of fresh medium. All samples were filtered (0.45 μm PTFE syringe filters, Isolab Laborgeräte Gmb, Eschau, Germany) and analyzed according to the described HPLC methods (Section 2.3.7).

#### 2.3.9. In Vitro Determination of Fe-Chelating Activity of Deferiprone Included in Nanocomposite Microparticles

The ability of DFP to bind free iron ions was determined against the iron indicator ferrozine by the method suggested by Dinis et al. [22]. Ferrozine (3-(2-pyridyl)-5,6-diphenyl-1,2,4-triazine p,p’-disulfonic acid) forms a stable complex with free iron ions (Fe(II)) with a characteristic red color and an absorption maximum at 562 nm. Compounds that bind Fe(II) reduce the amount of free Fe(II) in the solution and lower the concentration of the ferrozine-Fe(II) complex, which also leads to a decrease in the absorbance at 562 nm.

For the test, DFP solution in purified water with a concentration of 1 mg/mL, 0.1 mM FeSO_4_ solution, and 0.25 mM ferrozine solution were prepared. Placebo (38 mg) and drug-loaded (amount equivalent to 2 mg DFP) nanocomposite microspheres were dispersed in 2 mL purified water. Two milliliters of FeSO_4_ aqueous solution (0.1 mM) were added, the mixture was shaken, and the reaction was initiated by adding 2 mL of 0.25 mM ferrozine solution. A mixture of 0.1 mM FeSO_4_ and 0.25 mM ferrozine (1:1) was used as a control. The resulting reaction mixtures were left at room temperature for 10 min then filtered (cellulose acetate, 0.45 μm), and their absorbance was read spectrophotometrically at a wavelength of 562 nm against a blank to exclude interference from the excipients. Fe-chelating activity (Ferrous ions chelating, FIC) was calculated by the following formula:$$\%FIC=\frac{A_{0}-A_{1}}{A_{0}}\times 100$$
where A_0_ is the absorbance of the control and A_1_ is the absorbance of the test substances.

#### 2.3.10. Determination of the Ability of DFP-Loaded Nanoparticles to Bind Human Serum Albumin (HSA)

The binding ability of the nanoparticles to human serum albumin was investigated by fluorescence spectroscopy (FS). It is a sensitive method for studying the interaction between drug molecules and HSA, based on the characteristic fluorescent properties of HSA due to three amino acid residues in its structure: tryptophan, tyrosine, and phenylalanine. Upon excitation in the range from 280 nm to 295 nm, albumin emits intensely with a major emission peak at about 340 nm [23]. To investigate the potential interaction between HSA and the model nanoparticles, the fluorescence spectra of albumin were recorded in the absence and presence of increasing amounts of the nanocomposite microspheres. In the case of interaction, the fluorescence intensity of HSA gradually decreases with increasing amounts of nanoparticles [24].

The experiment was carried out at a temperature of 37 °C. Six samples were prepared for the analysis. The nanocomposite microspheres were dispersed in 1.1 mL saline. HSA solution with a concentration of 2.7 mg/mL was prepared. To each sample containing 1 mL of HSA solution, 20, 40, 60, 80, and 100 µL of the nanosuspension were added, respectively, and the samples were diluted with saline to a final volume of 1.1 mL. The samples were incubated for 1 h at 25 °C or 37 °C, after which the fluorescence analysis was performed. The spectra were recorded in the range of 310 ÷ 500 nm.

#### 2.3.11. Mucoadhesive Ability of the Nanocomposite Microspheres

The mucoadhesive ability of the nanocomposite microspheres was investigated by determining the maximum adhesive force applied to a unit area of freshly isolated sheep nasal mucosa (Figure 3) [25,26,27]. Within one hour after the death of the animal, the nasal mucosa was excised and fixed immobile on a paraffin disc. PBS buffer (pH 7.4) was used as a solution for storing and moistening the mucosa. A cylinder, obtained by compressing microspheres using a hydraulic press, was fixed on a microscopic coverslip. The sample was brought into contact with the mucosal surface tempered at 30 °C ± 0.5 °C for different time intervals (1, 3, 5, 10, 15, and 25 min). The maximum adhesive force applied per unit area was estimated by an interface system, composed of an Isometric force transducer TRI201 (LSI LETICA Scientific Instruments, Panlab S.L., Barcelona, Spain), an amplifier with very high gain, and a filtering system. The voltage obtained at the output of the amplifier was converted to digital form by a microcontroller-operated 13-bit analog-to-digital converter.

##### Statistical Analysis and Processing of Experimental Data from the Mucoadhesive Test and Fe-Chelating Activity

Statistical analysis was performed using GraphPad Prism v6.01 (GraphPad, Soft, Boston, MA, USA). To determine the statistically significant difference between the average values in the groups, a significance level of *p* < 0.05 was adopted. The data analyzed included results with statistically insignificant standard deviations from the mean, as demonstrated by Friedman’s one-factor test, Friedman ANOVA.

## 3. Results and Discussion

### 3.1. Preparation of Poly-ε-Caprolactone Nanoparticles by Double Emulsion/Solvent Evaporation Technique

The common emulsion solvent evaporation technique by forming an O/W emulsion type was not considered suitable for the inclusion of the water-soluble compound DFP due to its rapid distribution in the external aqueous phase. Therefore, based on the high-water solubility of DFP, a double emulsion technique was used to obtain nanoparticles, and a W/O/W emulsion type was chosen. DCM was used as the oil phase. The conditions of the applied technique (stirring speed, concentration of emulsifying agents, volume of internal and external aqueous phases) were established in preliminary experimental work, in which PCL 14 kDa at a concentration of 0.5% was used (Table 1). At a higher concentration of the polymer, particles with a size outside the desired range were obtained.

In the preliminary experimental work of our team, it was found that increasing the stirring speed from 11,000 rpm to 16,000 rpm during the first step of emulsification did not significantly affect the final particle size. The size decreased when the stirring speed was further increased to 21,000 rpm, at which speed particles of about 270 nm were obtained. During the second emulsification stage, increasing the stirring speed resulted in a proportional decrease in particle size from 865 nm to 274 nm. This shows that the stirring speed during the second step has a key effect on the particles’ size. This hypothesis was also confirmed by Yang et al., who found that in the double emulsion technique, the stirring speed during the second step is a determining factor for the particles’ size [28]. However, the stirring speed should not be too high, because instead of multiple, simple emulsion can be obtained [29]. The type and concentration of the emulsifier are important for the stability of the double emulsion. In preliminary tests, the inclusion of a lipophilic and a hydrophilic emulsifier, respectively, in the oil and outer water phases (W2) prevented droplet coalescence during the emulsification stages. One of the samples was prepared without an emulsifier in the oil phase, resulting in a rapid phase separation and inability to form a stable emulsion. Excessive amounts of lipophilic emulsifier may result in the formation of a simple W/O emulsion. A low concentration of the emulsifier in W2 led to an increase in the size of the resulting particles, probably due to the coalescence of the polymer droplets. Conversely, an excessive amount of hydrophilic emulsifier can solubilize the lipophilic phase and accelerate the movement of molecules from the internal to the external aqueous phase. Therefore, the concentration of the hydrophilic emulsifier should be as low as possible [30].

Varying the volume of the internal aqueous phase (W1) did not affect the final size of the particles formed after evaporation of the organic phase, while an increase in the volume of the external aqueous phase (W2) reduced the size of the particles. Larger particles were obtained at a W2 volume of up to 20 mL, and as the volume increased to 50 mL, the particle size decreased. This may be due to a decrease in the viscosity of the emulsion because of the larger volume of the external aqueous phase.

Based on the results of the preliminary experimental work, optimal conditions for obtaining PCL nanoparticles using the double emulsion technique were established (Table 2). Blank nanoparticles from PCL 80 kDa were also obtained under the indicated optimal conditions. Increasing the molecular weight of the polymer expectedly led to some increase in the average particle size, but it remained within the desired range for nose-to-brain delivery.

### 3.2. Preparation and Characterization of Deferiprone-Loaded Poly-ε-Caprolactone Nanoparticles

Two batches of DFP-loaded PCL nanoparticles were developed. The molecular weight of the polymer was varied to determine its influence on drug entrapment efficiency. Low (14,000 g/mol) and high (80,000 g/mol) molecular weight PCLs were used. The nanoparticles’ composition is presented in Table 3.

The nanoparticles from both batches had a monomodal size distribution (Figure 4) and relatively high ζ-potential values. This implies the presence of electrostatic repulsion between the particles, which prevents their aggregation and stabilizes the dispersion. The average particle size ranged from 213 ± 56 nm to 241 ± 91 nm (Table 3).

#### 3.2.1. Drug Loading and Entrapment Efficiency

The water solubility of DFP is strongly affected by pH. It is easily dissolved in polar solutions with acidic pH, but the solubility decreases sharply in a neutral medium. In our preliminary studies, low values for DL and DEE were estimated when purified water with a pH of about 6.0 was used for the preparation of the double emulsion (Table 4). The main disadvantage of the double emulsion technique is the diffusion of the hydrophilic active molecules into the dispersion medium during emulsification. When the goal is to develop a drug-delivery system based on particle design, high drug entrapment is a priority. Hence, the main task was to establish optimal conditions favoring the preferential distribution of the drug in the internal phase of the emulsion. Due to the pH-dependent solubility of DFP, a change in the pH of W1 and W2 was made to increase the amount of substance incorporated into nanoparticles. PBS (pH 2.6) was used for W1, in which DFP has high solubility, and PBS (pH 7.4) was used for W2, in which DFP is less soluble. Changing the pH resulted in an increase in DEE for both batches, as with the batch NP2-DFP prepared with a high molecular weight polymer, DEE was lower (17.15 ± 0.80%), while in the batch NP1-DFP obtained with PCL 14 kDa, DEE was 22.00 ± 2.16%. The lower affinity of hydrophilic molecules such as DFP to the hydrophobic PCL may be the reason for the low encapsulation efficiency. Alex et al. also reported a low encapsulation efficiency of the hydrophilic drug carboplatin in PCL nanoparticles in their work, prepared by double emulsion technique (DEE 27.95 ± 4.21%) [31].

#### 3.2.2. Nanoparticles Shape and Surface Morphology

Figure 5 and Figure 6 show SEM and TEM micrographs of the DFP-loaded NPs. Nanoparticles were spherical in shape with a smooth surface and uniform size distribution. SEM analysis confirms the DLS average diameter of the formulated nanoparticles.

#### 3.2.3. In Vitro Drug Release

The dissolution profiles of DFP from the two PCL-nanoparticle batches are shown in Figure 7. PCL has gained considerable attention as a drug delivery polymer due to its biocompatibility and slow degradation [32,33]. The release retarding effect of the hydrophobic PCL is due to the low penetration of water into the polymer matrix [34]. This effect has been used to control the fast drug release from different particulate dosage forms [35]. It was found that during the first 3 h of the study, about 30% of the drug was released. A reason for this could be the accumulation of the drug at the periphery of the nanoparticles during the intense evaporation of the organic phase [36]. After 3 h, there was a delay in the release, and by the 9th hour, only an additional 10% of the drug was released. Probably, a highly concentrated core of the NPs was formed, which is responsible for the incomplete release [37]. This was due to the relatively slow penetration of water in the PCL matrix. At the end of the 24 h study, about 60% from batch NP1-DFP and about 80% from batch NP2-DFP were released. The release profiles of DFP from the two batches are similar with minor differences attributed to the small variations in drug loading.

To date, only one DFP controlled-release oral dosage form has been developed. Binary matrix tablets containing DFP and thermoplastic polymers (Carbopol 974P and 971P) were prepared by an ultrasound-assisted tableting press. Prolonged release from hydrophilic matrices was successfully achieved, with 50% of the active substance released in 6 h [38]. The inclusion of DFP in PCL NPs leads to a delay in its release without observing a burst effect, which suggests that the drug remains unreleased at the site of administration (nasal cavity) and can be transported to the target site (brain tissue). However, further studies are required to prove this hypothesis.

### 3.3. Preparation and Characterization of Deferiprone-Loaded Nanocomposite Microspheres

#### 3.3.1. Optimization of the Microspheres’ Preparation

Sodium alginate was selected as the polymer carrier for microencapsulation of the nanoparticles. This natural polysaccharide has good mucoadhesive properties, making alginate microparticles suitable for application on mucous membranes such as the nasal mucosa [39]. Additionally, its biodegradability and hydrophilicity ensure its rapid dissolution in nasal secretions after application, which is essential for the transport of the drug-loaded nanostructures incorporated in the composite systems [13,40].

Spray drying was used to formulate the alginate composites. The technological parameters that influence the characteristics of the obtained microparticles are: the concentration of the starting material for spraying (polymer and drug substance); the rate at which the peristaltic pump feeds the atomizing nozzle; the amount of compressed gas required to disperse the sample in the form of droplets; the temperature of the hot gas used for the drying process; the rate at which the drying air is aspirated [41]. Although spray drying of sodium alginate solutions has been widely applied and proven as an effective microencapsulating method [42], the inclusion of nanoparticles, composed of a water-insoluble polymer such as PCL, can greatly modify the properties of the drying material and may require precise adjusting of the process conditions. Challenges in the technique used were to achieve a high production yield of microparticles, since the method is generally characterized by a great loss of material during pulverization [43]. Considering the nasal route of administration, it was crucial to avoid obtaining particles of very small sizes that could be inhaled into the lungs [21]. To achieve optimal production yield and particle size, spray drying parameters were adjusted within the limits outlined in the literature.

It was estimated that a drying temperature of 140 °C or lower was not effective enough to evaporate the aqueous phase and dry out the droplets sprayed through the nozzle. Consequently, the moist particles adhere to the equipment and cannot be collected as a final powder product. This led to a low yield of microparticles. An inlet temperature of 160 °C was evaluated as optimal for the particle yield. As expected, raising the inlet temperature resulted in a drier product with a higher yield of microparticles. Additionally, at the higher temperature, the obtained particles were larger in size. Although the contact with the hot air during the spray drying process was short, high temperatures are generally not recommended due to the risk to the stability of the drug and the polymers used [43]. In order to ensure the stability of the DFP and the encapsulating polymers, the cyclone of the spray dryer was jacketed with a cooling agent and insulated to maintain a lower temperature on the inner walls, where the dried powder was collected, thus maintaining the outlet temperature below 60 °C.

Another factor that has a strong impact on the yield and the size of microparticles is the gas flow rate. When more gas is passed through, smaller droplets are sprayed through the nozzle, resulting in smaller final dry particles [41,44]. At a rate of 400 L/h, the larger droplets may not dry efficiently, leading to a lower yield. On the other hand, at a rate of 600 L/h, the average particle size became too small. The optimal flow rate was found to be 500 L/h, as it resulted in satisfactory particle size and the highest yield. Producing very small particles can reduce the yield, as some of them are expelled during the process with the aspiration flow.

Two feed rates of the peristaltic pump were studied: 5 mL/min and 10 mL/min. A higher feed rate generally results in larger droplets/particles due to the larger amount of material sprayed. The pump speed should be restricted when the particles are not dry enough [45]. This was confirmed by the results obtained. At a rate of 10 mL/min, the larger droplets formed did not dry sufficiently and an unsatisfactory yield was estimated. This required the speed to be reduced to 5 mL/min, which did not greatly affect the average size of the obtained microparticles.

The microparticles were initially prepared using a 0.5% *w*/*v* sodium alginate solution. However, due to small particle size and unsatisfactory yield, the polymer concentration was increased to 1%. The higher polymer concentration led to a larger particle size and a higher yield because there was less liquid to evaporate and a more solid phase to form the powder. However, very high polymer concentrations can lead to challenges in atomization due to the high viscosity and can also reduce drug loading in the particles due to changes in the drug substance and polymer carrier ratio [46].

Based on the study of spray drying parameters, the following conditions were derived for obtaining nanocomposite microparticles from alginate: drying temperature 160 °C, gas flow rate 500 L/h, feed rate 5 mL/min, polymer concentration 1% *v*/*w*.

#### 3.3.2. Characterization of Nanocomposite Microspheres

The average diameter of the nanocomposite microspheres was 6.05 ± 1.78 µm (Table 5). The morphology of particles was observed via SEM analysis. Spray-dried composite structures were spherical in shape and had a smooth surface (Figure 8). Similar observations were also reported by Bagheri et al. [47]. For intranasally administered powders, optimization of particle size and morphology is necessary to minimize the risk of irritation of the nasal mucosa and to improve powder flowability and nasal deposition [48,49,50]. The spherical shape is expected to facilitate the delivery of the preparation by nasal devices [45]. High yields of about 86.93% ± 3.04% were obtained, which are considered optimal, taking into account the large losses typical of the production method at a laboratory scale [43,51].

The particle size distribution of the composite microparticles is presented in Figure 9. A monomodal distribution was observed with a predominance of a fraction of particles with sizes between 2 and 20 µm. A fraction of smaller particle sizes (0.1–1.0 µm) was also observed, which was expected since spray drying is a method that produces microparticles in a relatively wide size range. The size range of the nanocomposite microspheres corresponds to that of nasal powders produced by spray drying [52,53,54].

#### 3.3.3. In Vitro Drug Release

The release of DFP from nanoparticles was prolonged with only 60% of the drug released in 24 h (Figure 7). The inclusion of nanoparticles in the composite structure led to even further delays in the release of DFP (Figure 10). During the first 3 h, around 10% were released from the composite structure, while at the same time, twice the amount of drug was released from nanoparticles. A plausible explanation is the dissolution of sodium alginate in the release medium, which increases the viscosity and slows down the dissolution and diffusion of DFP. Similar conclusions were reported by other authors, using sodium alginate as a mucoadhesive carrier in nasal gels [55], buccal patches [56], and rectal suppositories [57]. The type of dosage form also influences the drug release rate; for example, alginate was used as a carrier for salbutamol sulfate in sublingual films [58] and tablets [59]. The drug release was found to be prolonged for the films [58] and rapid for the tablets [59].

#### 3.3.4. FTIR Spectroscopy

The infrared spectra of DFP, PCL, sodium alginate, and DFP-loaded nanocomposite microspheres (M-NP1-DFP) are shown in Figure 11. A peak is registered in the spectrum of DFP at 1629 cm^−1^ characteristic of C=O group. A band is observed at 1380 cm^−1^, which together with the band at 1463 cm^−1^ is due to -CH3 groups in the molecule. The characteristic peak of C=C bonds is 1513 cm^−1^, and the presence of C–N bonds in the molecule is confirmed by the peak at 1030 cm^−1^ [60]. In the spectra of PCL, a doublet at 2946 cm^−1^ and 2867 cm^−1^ is observed due to symmetric and asymmetric vibrations of the aliphatic C–H bonds as well as a characteristic band for C=O at 1721 cm^−1^. The valence vibrations of C-C and C-O bonds in the PCL molecule appear at 1293 cm^−1^ [61]. A broad characteristic peak at 3500 cm^−1^ is observed in the spectrum of sodium alginate, which is due to the vibration of the large number of -OH groups in the molecule. Two characteristic peaks at 1603 cm^−1^ and 1411 cm^−1^ are distinguished due to symmetric and asymmetric vibrations of the -COO groups [62]. In the infrared spectrum of batch M-NP1-DFP, the peak at 3253 cm^−1^ confirms the presence of -OH groups. The peaks at 1724 cm^−1^, 1293 cm^−1^, and 1241 cm^−1^ correspond to the indicated characteristic peaks for PCL. The peak at 1629 cm^−1^ corresponding to the C=O group in the DFP molecule is shifted at 1609 cm^−1^, which may be due to overlap with the -COOH groups of alginates. In the ATR spectrum of the composite structure, the remaining characteristic peaks of DFP are also present without significant changes. No new peaks are observed in the ATR spectra of the model system, which shows that during their preparation, no new chemical bonds are formed because of the interaction between the active substances and the polymers.

#### 3.3.5. Differential Scanning Calorimetry (DSC)

The physical state of DFP incorporated into nanocomposite microspheres was analyzed using differential scanning calorimetry (DSC). The resulting thermograms are presented in Figure 12 and Table 6. The DSC thermogram of sodium alginate shows an endothermic peak at 111 °C and an exothermic peak at 243 °C. The first peak corresponds to the evaporation of the included water molecules, while the exothermic indicates the oxidative degradation of the polymer (Figure 12A). The thermogram of PCL demonstrated a broad endothermic peak at about 76 °C, which corresponds to the melting temperature of the polymer (Figure 12B). Similar endothermic peaks corresponding to the melting point were also observed in the DFP (284 °C) thermograms (Figure 12C). The thermogram of the microstructures of model M-NP1-DFP contains two endothermic peaks: one at 60 °C, which corresponds to the melting point of PCL and confirms the incorporation of the polymer nanoparticles into the composite, and a second endothermic peak at 165 °C, which is due to the Span 85 used in the preparation of the multiple emulsion (Figure 12D). The exothermic peak at 252 °C is due to the DFP included in the nanoparticles. In the DSC thermogram of the placebo nanocomposite microspheres (M-NP-placebo), one endothermic peak was registered at 60 °C, which is due to the PCL placebo nanoparticles included in the composite structure. The second peak, which is exothermic and recorded at 248 °C, corresponds to the thermal decomposition of the structure-forming polymer, sodium alginate (Figure 12E).

#### 3.3.6. Mucoadhesive Ability

When developing a carrier for nasal administration, its mucoadhesive ability should be evaluated. Mucociliary clearance is the main obstacle for better contact between the carrier and the nasal mucosa and, accordingly, for a higher rate of drug absorption [63,64]. Several methods have been developed to study mucoadhesion [65,66]. There is no standard apparatus for determining adhesive strength and there is a great variety of potential techniques [67]. In the present work, the mucoadhesive ability of nanocomposite microspheres as a drug delivery system was investigated ex vivo in direct contact with the nasal mucosa [25,26]. The maximum force required to detach a cylinder of compressed nanocomposite microparticles from native sheep nasal mucosa after different contact times (1, 3, 5, 10, 15, and 25 min) was determined.

The results of the study (Figure 13) showed a high degree of attachment to the nasal mucosa. The chronological change in adhesion strength showed a statistically significant difference at all time points compared to the initial moment. This is an indicator of achieving a high degree of attachment even after one minute of contact with the nasal mucosa. As the contact time increases, the maximum adhesive strength increases and reaches a maximum after five minutes of contact. It was found that a contact time longer than 5 min did not lead to an additional increase in the adhesive strength. On the contrary, a decrease and reaching a stable mechanical tension was registered after 15 min.

#### 3.3.7. In Vitro Determination of Fe-Chelating Activity

The study aims to determine whether there is a change in the chelating ability of DFP after its incorporation into the composite structures. The test is based on the fact that in the presence of iron ions, ferrozine forms a complex compound (ferrozine—Fe(II)) with a characteristic red color, which has an absorption maximum of 562 nm. In the presence of other chelating agents in the solution, the formation of the complex is prevented, which leads to a decrease in the absorption. In a water solution, DFP prevented the formation of the ferrozine—Fe(II) complex and demonstrated chelating activity of over 95% (Figure 14). The inclusion of DFP in the nanocomposite microspheres led to a slight decrease in Fe-chelating activity (84.38%), but that was probably due to the incomplete and delayed release of DFP from the composite structures. The placebo nanocomposite microspheres (M-NP-Placebo) also demonstrated some Fe-chelating activity (45.2%) resulting from the formation of a complex between alginate and iron ions. The mechanism of this interaction has been studied for a long time [68].

#### 3.3.8. Determination of the Binding Ability of Nanoparticles Loaded with DFP to Human Serum Albumin (HSA)

Albumin is the major protein in plasma and the cerebrospinal fluid. It is a pliable molecule with the ability to change its structure depending on environmental conditions such as temperature, pH, or ionic strength. It can bind many different molecules and can even act as an antioxidant [69].

Proteins such as HSA, lipoproteins, and globulin can be adsorbed on the surface of drug-loaded nanoparticles, thereby altering the in vivo release behavior and targeting sites of the nanoparticles [70,71,72].

In the present research, fluorescence spectroscopy was used for in situ investigation of nanoparticle–protein interactions (Figure 15). Quantitative analysis of HSA binding was performed using the reduction of fluorescence emission of albumin with a characteristic peak at about 340 nm. The performed analysis showed a noticeable decrease in fluorescence intensity after the addition of nanosuspension from batch NP1-DFP (Figure 15).

DFP-loaded nanoparticles were found to interact with HSA. A fluorescence study of the HSA solution with placebo nanoparticles was also performed. A negligible reduction in fluorescence was registered compared to that observed in batch M-NP1-DFP when increasing the amount of the ligand (Figure 13).

These results reveal a minor interaction of the nanosized carrier with albumin and suggest a preferential interaction of the drug substance DFP with the protein. It is logical to conclude that this effect was due to the interaction between HSA and the drug molecules. However, considering the low release rate of DFP from the nanocarriers (<10%, Figure 6) within 1 h (the duration of the analysis), it can be assumed that the observed decrease in fluorescence spectra is not solely due to the interaction between the active substance with albumin. It can be suggested that the incorporation of DFP into PCL nanoparticles significantly changed their surface properties and geometry because their affinity for binding to the protein increased dramatically.

## 4. Conclusions

In this study, nanocomposite microspheres loaded with DFP were successfully produced by emulsification solvent evaporation method and spray drying. The resulting particles have high mucoadhesive ability and an average size of 6.05 ± 1.78 µm. pH of the aqueous phases influenced the amount of drug incorporated into the nanoparticles. The drug release study confirmed the feasibility of the developed carrier for prolonged delivery of DFP. The inclusion of DFP in PCL nanoparticles and the subsequent incorporation of nanoparticles into composite microspheres did not lead to a change in its Fe-chelating activity. The binding affinity of PCL nanoparticles to human serum albumin was determined. In conclusion, it can be summarized that the loading of polymer nanoparticles with active ingredients is crucial for their behavior, possible bio-nano interactions, and thus their potential application to achieve specific therapeutic goals (targeted delivery, controlled release, etc.).

## Figures and Tables

**Figure 1 jfb-15-00329-f001:**
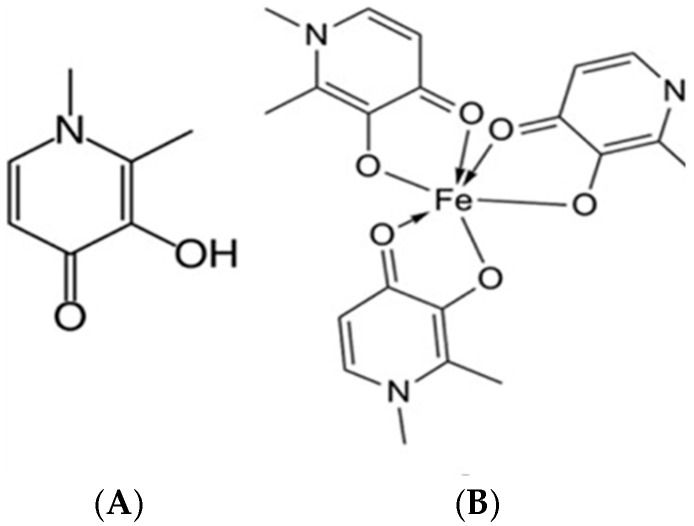
Chemical structure of DFP (**A**) and iron ion binding sites (**B**).

**Figure 2 jfb-15-00329-f002:**
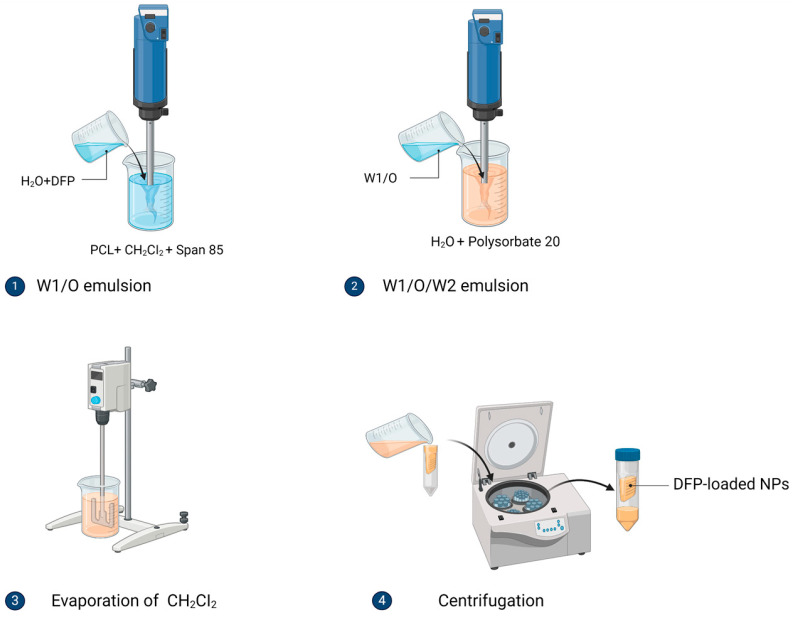
Schematic illustration of nanoparticle preparation by double emulsion technique (figure created with BioRender.com).

**Figure 3 jfb-15-00329-f003:**
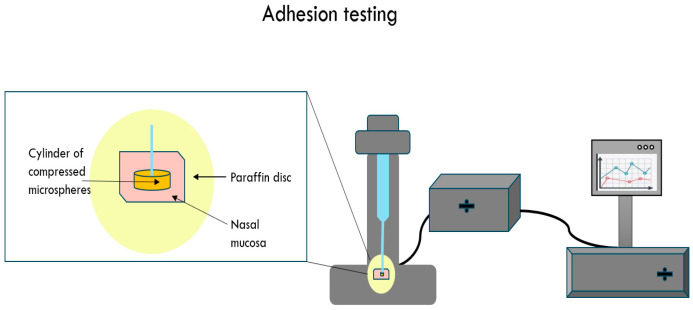
Schematic illustration of the experimental set-up for the study of mucoadhesive ability.

**Figure 4 jfb-15-00329-f004:**
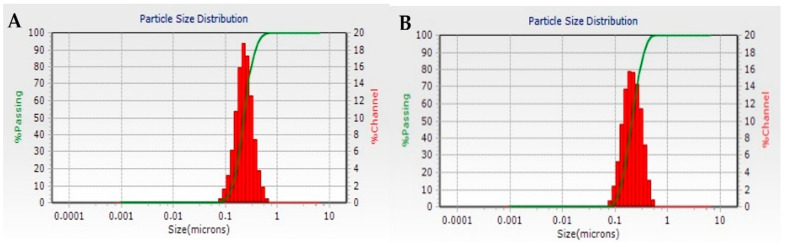
DLS histograms of nanoparticles from batches NP1-DFP (**A**) and NP2-DFP (**B**).

**Figure 5 jfb-15-00329-f005:**
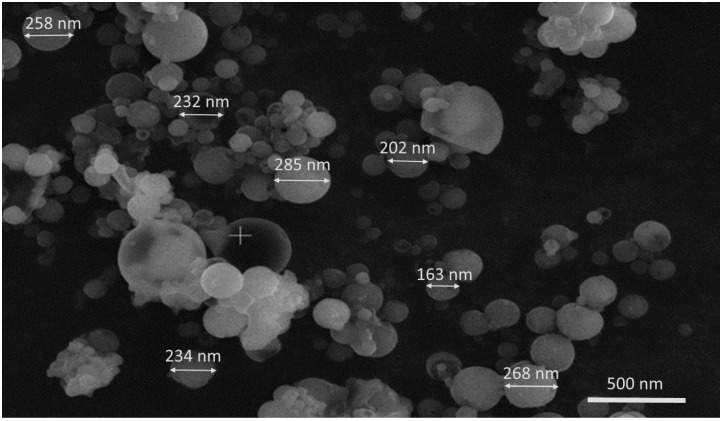
SEM micrograph of DFP-loaded nanoparticles from model NP1-DFP (20,000×).

**Figure 6 jfb-15-00329-f006:**
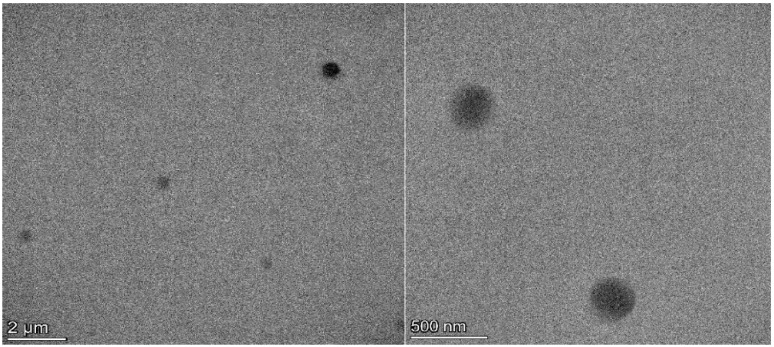
TEM micrographs of nanoparticles loaded with DFP from model NP1-DFP (28,000×).

**Figure 7 jfb-15-00329-f007:**
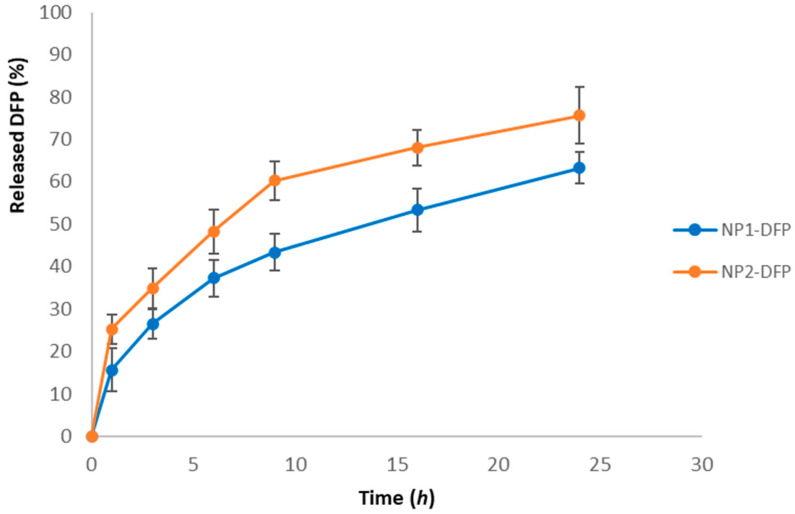
DFP release profiles from batches NP1-DFP and NP2-DFP.

**Figure 8 jfb-15-00329-f008:**
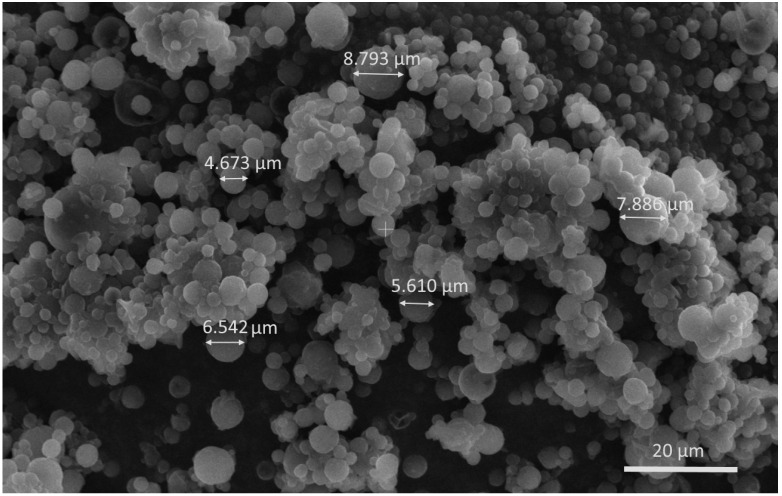
SEM micrograph of nanocomposite microspheres from batch M-NP1-DFP (3500×).

**Figure 9 jfb-15-00329-f009:**
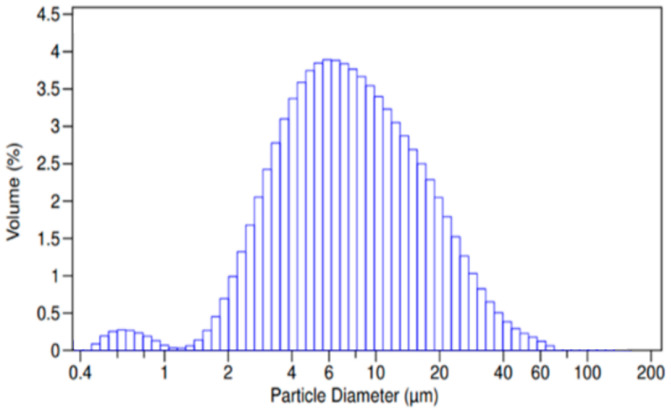
Particle size distribution of batch M-NP1-DFP.

**Figure 10 jfb-15-00329-f010:**
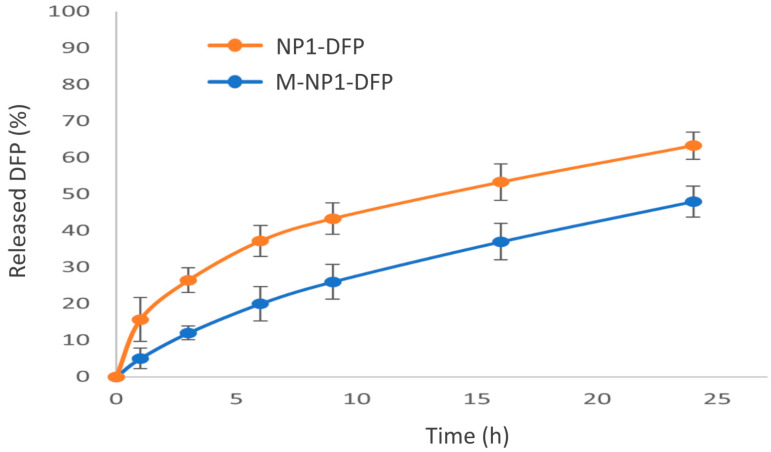
Release profiles of DFP from nanoparticles (NP1-DFP) and composite microspheres (M-NP1-DFP).

**Figure 11 jfb-15-00329-f011:**
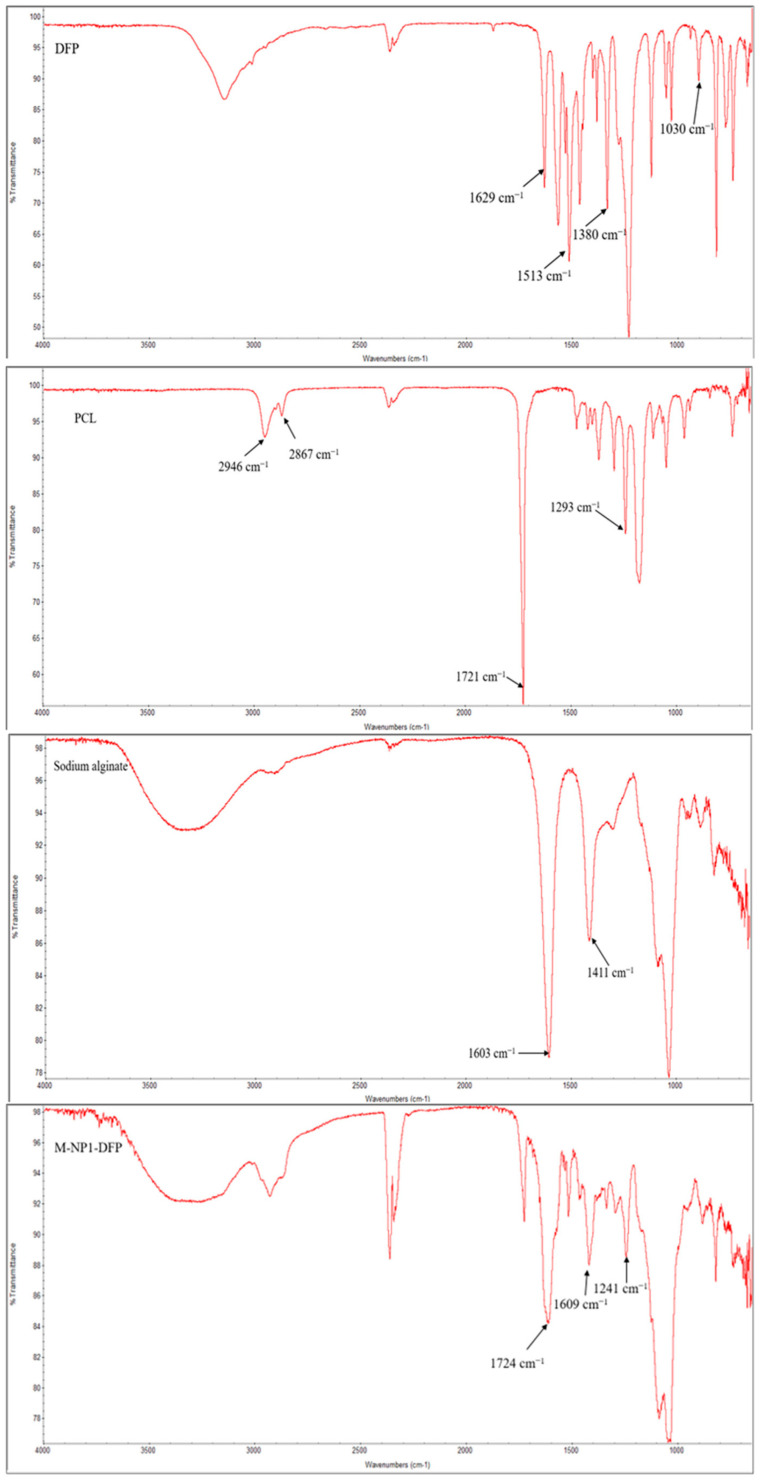
FTIR spectra of DFP, PCL, sodium alginate, and nanocomposite microparticles of model M-NP1-DFP.

**Figure 12 jfb-15-00329-f012:**
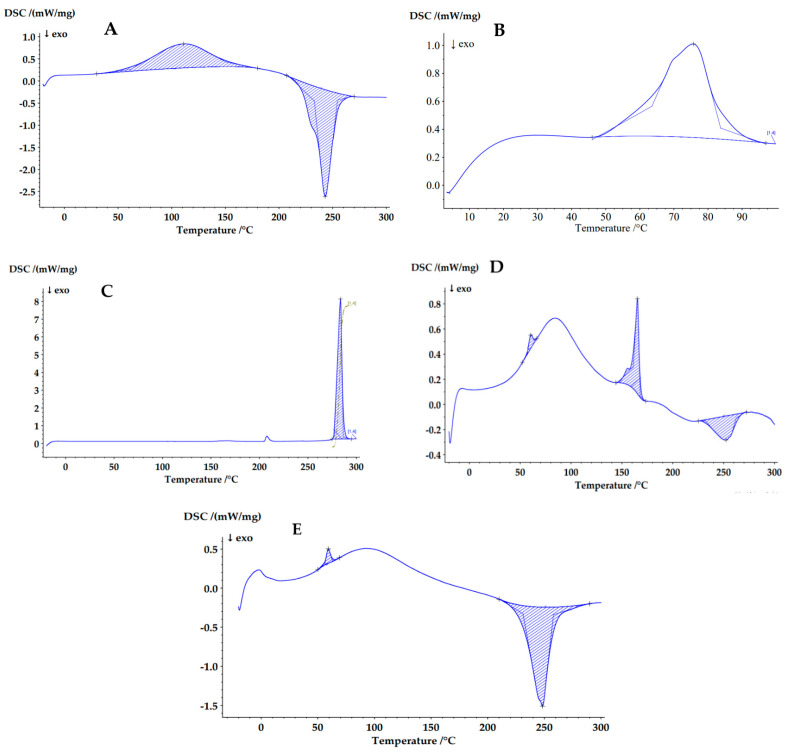
DSC thermograms of sodium alginate (**A**), PCL (**B**), DFP (**C**), and particles from batches M-NP1-DFP (**D**) and M-NP-Placebo (**E**).

**Figure 13 jfb-15-00329-f013:**
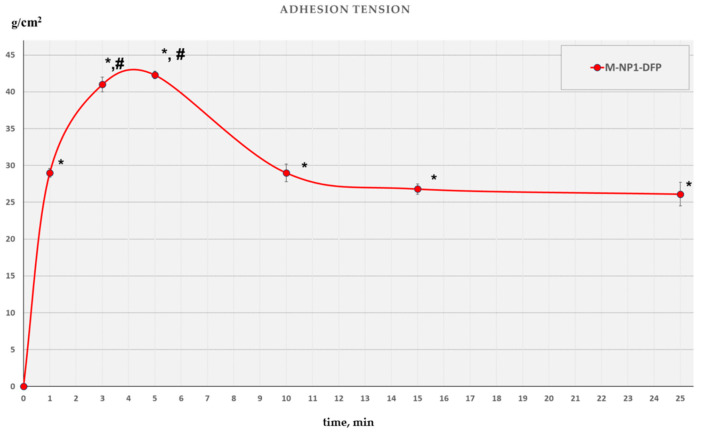
Chronological change of the maximal adhesion force in batch M-NP1-DFP. The symbols *, #, indicate the presence of a statistically significant difference compared to the initial moment (*) and after the established steady state of the process 15 min (#).

**Figure 14 jfb-15-00329-f014:**
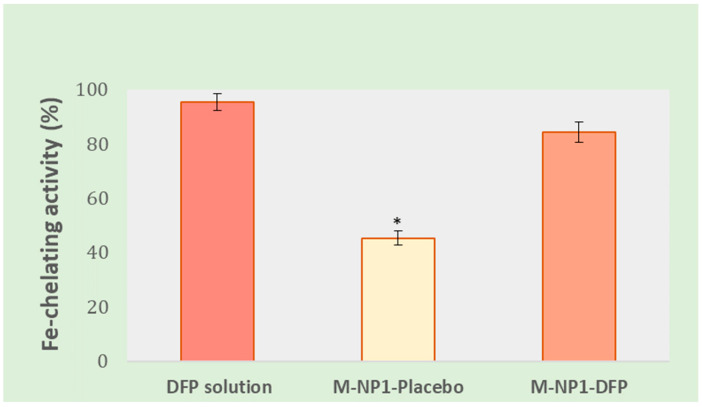
Comparison of Fe-chelating activity of DFP solution, nanocomposite microspheres without the drug (M-NP-Placebo), and DFP-loaded nanocomposite microspheres (M-NP1-DFP). Data is presented as mean values ± SD, *n* = 3. * Indicates a statistically significant result (*p* < 0.05).

**Figure 15 jfb-15-00329-f015:**
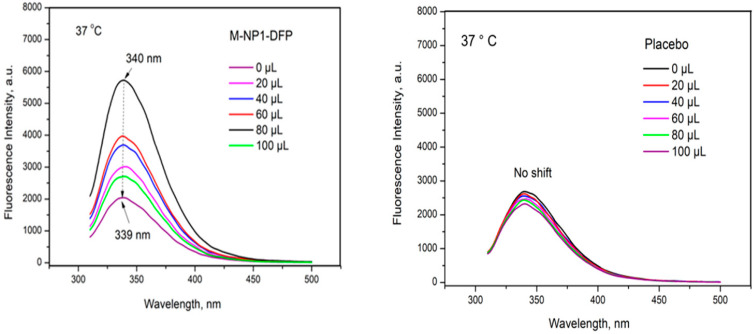
Fluorescence spectra of HSA (2.7 mg/mL) upon addition of different volumes of samples containing dispersed particles from batch M-NP1-DFP and placebo particles at 37 °C.

**Table 1 jfb-15-00329-t001:** Data from preliminary experimental work. “**Bold**” values indicate changed parameters.

Conditions	Primary Emulsion’s Parameters (W1/O)			Double Emulsion’s Parameters W1/O/W2			Average Particle Size (nm) ± SD
	Inner Phase, W1 (mL)	Stirring Speed (rpm)	Span 85 (%)	Polysorbate 20(%)	Outer Phase, W2 (mL)	Stirring Speed (rpm)	
Stirring speed for W1/O	1.0	**11,000**	1.0	1.0	50.0	25,000	356 ± 65
	1.0	**16,000**	1.0	1.0	50.0	25,000	310 ± 25
	1.0	**21,000**	1.0	1.0	50.0	25,000	270 ± 21
Stirring speed for W1/O/W2	1.0	21,000	1.0	1.0	50.0	**16,000**	865 ± 18
	1.0	21,000	1.0	1.0	50.0	**21,000**	350 ± 32
	1.0	21,000	1.0	1.0	50.0	**25,000**	274 ±12
Concentration of Span 85	1.0	21,000	**-**	1.0	50.0	25,000	unstable W1/O
	1.0	21,000	**0.5**	1.0	50.0	25,000	480 ± 29
	1.0	21,000	**1.0**	1.0	50.0	25,000	290 ± 16
Concentration of Polysorbate 20	1.0	21,000	1.0	**0.5**	50.0	25,000	310 ± 27
	1.0	21,000	1.0	**1.0**	50.0	25,000	278 ± 10
	1.0	21,000	1.0	**1.5**	50.0	25 000	254 ± 8
Volume of W1	**1.0**	21,000	1.0	1.0	50.0	25,000	285 ± 11
	**1.5**	21,000	1.0	1.0	50.0	25,000	310 ± 7
	**2.0**	21,000	1.0	1.0	50.0	25,000	369 ± 13
Volume of W2	1.0	21,000	1.0	1.0	**20.0**	25,000	420 ± 21
	1.0	21,000	1.0	1.0	**30.0**	25,000	390 ± 19
	1.0	21,000	1.0	1.0	**50.0**	25,000	266 ± 8

**Table 2 jfb-15-00329-t002:** Optimal conditions for the preparation of nanoparticles by the double emulsion technique.

Primary Emulsion’s Parameters W1/O				Double Emulsion’s Parameters W1/O/W2		
Inner Phase, W1 (mL)	Stirring Speed (rpm)	*PCL* (%)	Span 85 (%)	Polysorbate 20 (%)	Outer Phase, W2 (mL)	Stirring Speed (rpm)
1.0	21,000	0.5	1.0	1.0	50.0	25,000

**Table 3 jfb-15-00329-t003:** Composition and characteristics of deferiprone-loaded nanoparticles (means ± SD, *n* = 3; “-” not included).

Sample Code	DFP (mg)	PCL (14 kDa) (mg)	PCL (80 kDa) (mg)	DFP: PCL	Mean Diameter (nm)	PDI	ζ-Potential (mV)
NP1-DFP	12.5	25.0	-	1:2	213 ± 56	0.820	−12.70 ± 0.5
NP2-DFP	12.5	-	25.0	1:2	241 ± 91	0.535	−18.22 ± 0.1

**Table 4 jfb-15-00329-t004:** Effect of pH of W1 (internal aqueous phase) and W2 (external aqueous phase) on entrapment efficiency (DEE), drug loading (DL) (mean values ± SD. *n* = 3).

Sample Code	pH W1	pH W2	DEE (%)	DL (%)
NP1-DFP	6.0	6.0	3.31 ± 0.97	1.62 ± 0.42
NP2-DFP	6.0	6.0	2.72 ± 0.12	1.34 ± 0.06
NP1-DFP	2.6	7.4	22.00 ± 2.16	9.08 ± 0.95
NP2-DFP	2.6	7.4	17.15 ± 0.80	7.40 ± 0.50

**Table 5 jfb-15-00329-t005:** Characteristics of DFP-loaded nanocomposite microspheres (means ± SD, *n* = 3); drug entrapment efficiency (DEE), drug loading (DL).

Mean Diameter (µm)	DEE (%)	DL (%)	Yield (%)
6.05 ± 1.78	36.28 ± 0.32	5.18 ± 0.06	86.93 ± 3.04

**Table 6 jfb-15-00329-t006:** DSC thermographic peaks of sodium alginate, PCL, DFP, and particles from batches M-NP1-DFP and M-NP-Placebo.

Sample	Area (J/g)	Peak (°C)	Onset (°C)	End (°C)	Width (°C)	Height (mW/mg)
Sodium alginate	200.0	111.0	61.6	150.2	69.3	0.543
	−257.2	243.0	232.9	254.5	21.6	2.414
poly-ε-caprolactone (PCL)	71.19	75.8	63.5	83.7	20.5	0.6681
Deferiprone (DFP)	262.4	283.7	277.2	286.5	6.4	7.881
Microcomposite particles (M-NP1-DFP)	3.1	60.2	56.8	63.7	5.3	0.1078
	29.93	165.2	161.2	168.0	6.2	0.757
	−24.39	252.4	232.4	262.2	24.7	0.1912
M-NP-Placebo	5.41	59.3	56.4	62.9	5.0	0.185
	−147.8	248.3	231.1	258.0	20.8	1.264

## Data Availability

The original contributions presented in the study are included in the article, further inquiries can be directed to the corresponding author.

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
