# Peer review of "Development of Nanocomposite Microspheres for Nasal Administration of Deferiprone in Neurodegenerative Disorders"

_jfb, 2024, doi:10.3390/jfb15110329_

Round 1
Reviewer 1 Report
Comments and Suggestions for Authors
The paper with title: “Development of Nanocomposite Microspheres for Nasal Ad-2 ministration of Deferiprone in Neurodegenerative Disorders”, presents in a clear way the synthesis, validation of the conditions, of the components in order to obtain sustainable systems for the administration of deferiprone embedded in nanoparticles. The study is well written with many analyzes and many clearly explained and argued conclusions. As a recommendation for improving the general aspect of the work, it would be that the data presented on figures 10(a-e) be entered in a table.
The work presents an original study with clear therapeutic applications.
In the article you claim that "To obtain a high production yield and an average particle diameter above 5 µm (adequate
for nasal administration) [15], the process parameters were varied as follows: drying temperature from 120 °C to 160 °C, feed flow rate 5÷7.5 mL/min and gas flow rate from 400 to 600 l/h ." - these conditions do not influence the stability of the components within the particles because all components except DFP show thermal degradation processes up to these temperatures? How can you explain the stability at such high temperatures?
Author Response
Thank you very much for taking the time to review this manuscript. Please find the detailed responses below and the corresponding revisions/corrections highlighted in the re-submitted files.
Comment 1: As a recommendation for improving the general aspect of the work, it would be that the data presented on figures 10(a-e) be entered in a table.
Response 1: Thank you for the recommendation! The data, presented on figures 10(a-e) was entered in a table (Table 6) as suggested.
Comment 2: In the article you claim that "To obtain a high production yield and an average particle diameter above 5 µm (adequate for nasal administration) [15], the process parameters were varied as follows: drying temperature from 120 °C to 160 °C, feed flow rate 5÷7.5 mL/min and gas flow rate from 400 to 600 l/h ." - these conditions do not influence the stability of the components within the particles because all components except DFP show thermal degradation processes up to these temperatures? How can you explain the stability at such high temperatures?
Response 2: Thank you for the question! Spray drying is a method suitable for microencapsulation even of heat-sensitive materials (cells, essential oils, etc.). It operates at high inlet temperature to evaporate the liquid phase and form dry particles, but the contact time of the sprayed droplet with the hot air is extremely short to jeopardize the stability of the material. On the other hand, the evaporating moisture does not allow the material to reach the inlet temperature used, even for this brief moment of contact. The temperature of the air leaving the drying chamber after the drying process (outlet temperature) is much lower than the inlet temperature due to the cooling effect of water evaporation from the spray droplets. It varied according to the used inlet temperature, but it was kept below 60 °C. Additionally in order to preserve the stability of the particles (especially of poly-ε-caprolactone) the cyclone of the spray dryer was jacketed with cooling agent and insulated to maintain a lower temperature on the inner walls where the dried powder was collected.
Reviewer 2 Report
Comments and Suggestions for Authors
The manuscript “Development of Nanocomposite Microspheres for Nasal Administration of Deferiprone in Neurodegenerative Disorders” is focused on formulating a drug delivery system for the treatment of elevated brain iron levels. The aim of the paper is to develop mucoadhesive deferiprone-loaded nanocomposite microspheres for nasal delivery. The article is well written and interesting for researchers in the field nasal drug delivery and pharmaceutical technology, however, there are a few points to be considered.
First, the manuscript lacks proper discussion. In the Results and Discussion section, the authors list their results, but the discussion of the results is missing or is insufficient (i. e. sections 3.3.2. – 3.3.8). That reflects the fact that there are only 41 references, which is a low number for such an elaborate research topic.
There are also several specific points to be considered:
- Lines 93 - 95: Please add the reference for this sentence.
- Lines 127-128: did the oil phase contain polymer solution, or should it be written "PCL" that was dissolved in the oil phase, as is stated in Figure 2?
- Section 2. 2. 2. Preparation of nanocomposite microspheres
o Since the addition of nanoparticles in polymer solution resulted in a dispersion, how was the homogeneity during the drying process ensured? Was the sampled stirred constantly?
o Please correct the division sign in line 155.
o How was the feed rate measured/calculated (line 155)?
- Lines 197-200: is this yield for the spray drying process? If so, W2 in the equation should include the amounts of the drug and all the excipients in the dispersion.
- Section 2.3.7. Estimation of drug loading (DL) and entrapment efficiency (EE) - drug loading and encapsulation efficiency should be calculated also for the prepared microspheres, and that method should be described.
- Line 225: Please explain what it means “from each model”.
- Line 283: Please elaborate which kind of an interface system was used for the experiment. Also, if possible, it would help if the authors could add a figure representing the mucoadhesion determination experiment.
- Tables 3 and 4: I would suggest the authors to combine Table 3 and Table 4 into just one table, i. e. just add mean diameter, PDI and zeta potential values in Table 3, and change the table title accordingly.
- Section 3.2.1. Drug loading and Entrapment efficiency, Table 5: please comment on the drug loading results. Is the obtained amount enough to achieve a therapeutic effect?
- Section 3.2.2. Nanoparticles shape and surface morphology: Please comment on the particle size observed by TEM and SEM compared to the DLS results.
- Section 3.2.3. In vitro drug release: The authors should add discussion regarding the drug release results, and put these results in literature context, which is currently missing.
- Lines 438 - 441: since the particles, including the drug and the polymer(s), never reach inlet drying temperature due to the gas heat being used for solvent evaporation, spray drying is a suitable technique even for some mildly thermosensitive substances. The temperature that is more important is outlet temperature. The authors should state the outlet temperature and comment on its influence on drug and polymer stability.
- Section 3.3.2.: Please expand the discussion regarding the microsphere characterisation and include references.
- Section 3.3.2.: the authors should determine microsphere particle size distribution, since it is a critical parameter that influences efficacy of nasal drug delivery. It is not enough to have a mean value and SD since guidelines state that it is crucial that the majority of particles for nasal drug delivery is larger than 10 μm.
- Figure 8. Please check the legend in the figure, the figure currently does not match the discussion. Maybe the samples are switched.
- The section 3.3.6. is also missing references. Furthermore, as mentioned in the comment for the Methods sections, it is not completely clear how the detachment force was measured. Please elaborate and compare the obtained results with literature.
- Lines 621 – 623: The authors state that the particle size of around 6 μm is appropriate for nasal administration, however, as mentioned earlier in the comments, the majority of particles need to be over 10 μm. The authors should correct this conclusion.
Author Response
Thank you very much for taking the time to review this manuscript. Please find the detailed responses below and the corresponding revisions/corrections highlighted in the re-submitted files.
Comment 1: First, the manuscript lacks proper discussion. In the Results and Discussion section, the authors list their results, but the discussion of the results is missing or is insufficient (i. e. sections 3.3.2. – 3.3.8). That reflects the fact that there are only 41 references, which is a low number for such an elaborate research topic.
Response 1: The discussion of the results was expended, and more references were added as suggested by the reviewer.
Comment 2: Lines 93 - 95: Please add the reference for this sentence.
Response 2: References were added.
Comment 3: Lines 127-128: did the oil phase contain polymer solution, or should it be written "PCL" that was dissolved in the oil phase, as is stated in Figure 2?
Response 3: PCL was dissolved in the oil phase, as it is stated in Figure 2. It was corrected in the text to make it clearer.
Comment 4: Since the addition of nanoparticles in polymer solution resulted in a dispersion, how was the homogeneity during the drying process ensured? Was the sampled stirred constantly?
Response 4: The dispersion was continuously stirred on a magnetic stirrer at 300 rpm to maintain homogeneity during the spray drying. It was added in the Materials and Methods section.
Comment 5: Please correct the division sign in line 155.
Response 5: The division sign was corrected.
Comment 6: How was the feed rate measured/calculated (line 155)?
Response 6: The peristaltic pump rate was varied from 15 to 25% (the pump rate in % is indicated by the apparatus). The pump rate from % was then converted into feed flow (mL/min), respectively 5-7.5 mL/min, using the diagram “Fig. 6.3: Pump settings versus throughput” provided in the Buchi Mini Spray Dryer B-290 Operation Manual (page 49).
Comment 7: Lines 197-200: is this yield for the spray drying process? If so, W2 in the equation should include the amounts of the drug and all the excipients in the dispersion.
Response 7: This equation is also used to calculate the yield of microparticles obtained by spray drying. W2 include the amount of polymers (PCL, sodium alginate) and the drug. It was clarified in the text.
Comment 8: Section 2.3.7. Estimation of drug loading (DL) and entrapment efficiency (EE) - drug loading and encapsulation efficiency should be calculated also for the prepared microspheres, and that method should be described.
Response 8: The method was added and described.
Comment 9: Line 225: Please explain what it means “from each model”.
Response 9: We prepared two batches of PCL nanoparticles loaded with DFP (NP1-DFP and NP2-DFP). The word “model” was replaced with “batch”.
Comment 10: Line 283: Please elaborate which kind of an interface system was used for the experiment. Also, if possible, it would help if the authors could add a figure representing the mucoadhesion determination experiment.
Response 10: The interface system is composed of Isometric force transducer TRI201 (LSI LETICA Scientific Instruments, Panlab S.L., Barcelona, Spain), amplifier with very high gain and parasitic filtering system signals. The voltage obtained at the output of the amplifier is converted to digital form by a 13bit analog-to-digital converter, implemented based on a programmable microcontroller. A figure of the experiment was added as suggested.
Comment 11: Tables 3 and 4: I would suggest the authors to combine Table 3 and Table 4 into just one table, i. e. just add mean diameter, PDI and zeta potential values in Table 3, and change the table title accordingly.
Response 11: Thank you for the suggestion. Tables were combined as recommended by the reviewer.
Comment 12: Section 3.2.1. Drug loading and Entrapment efficiency, Table 5: please comment on the drug loading results. Is the obtained amount enough to achieve a therapeutic effect?
Response 12: The oral dose of DFP is relatively high and it is taken at short time intervals due to its short plasma half-life. This is a reason for non-compliance with therapy especially in elderly patients. The therapeutic dose of DFP in patients with neurodegenerative disorders is still under investigation and debate (https://doi.org/10.1111/cns.14607). The nasal route of administration allows targeted delivery of the drug to brain tissue (through nose-to-brain delivery mechanisms), which significantly reduces systemic exposure, rapid metabolism and suggests that a much lower dose would be required to observe the desired therapeutic effect. Recent clinical trial has demonstrated that short term comparatively low oral dose DFP therapy in Parkinson’s disease subjects is associated with decreases of iron in specific brain regions (https://doi.org/10.1038/s41598-017-01402-). Considering the quantity of powder that can be administered per nostril per shot (about 10–25 mg ), the obtained drug content within the microspheres may be sufficient to enable nasal administration of DFP therapeutic dose. Indeed, future longer-term studies are necessary to assess the neuroprotective dose of DFP after nasal administration.
Comment 13: Section 3.2.2. Nanoparticles shape and surface morphology: Please comment on the particle size observed by TEM and SEM compared to the DLS results.
Response 13: SEM micrographs of the DFP-loaded nanoparticles and microparticles were revised providing information about the size of the individual particles. SEM analysis confirms the DLS average diameter of the formulated nanoparticles and this conclusion was added in the manuscript as suggested by the reviewer.
Comment 14: Section 3.2.3. In vitro drug release: The authors should add discussion regarding the drug release results, and put these results in literature context, which is currently missing.
Response 14: The discussion was expanded as suggested by the reviewer.
Comment 15: Lines 438 - 441: since the particles, including the drug and the polymer(s), never reach inlet drying temperature due to the gas heat being used for solvent evaporation, spray drying is a suitable technique even for some mildly thermosensitive substances. The temperature that is more important is outlet temperature. The authors should state the outlet temperature and comment on its influence on drug and polymer stability.
Response 15: The outlet temperature varied according to the used inlet temperature, but it was kept below 60 °C. In order to ensure the stability of the particles (especially of poly-ε-caprolactone) the cyclone of the spray dryer was jacketed with cooling agent and insulated to maintain a lower temperature on the inner walls where the dried powder was collected. It was stated in the manuscript as suggested by the reviewer.
Comment 16: Section 3.3.2.: Please expand the discussion regarding the microsphere characterisation and include references.
Response 16: The discussion was expanded, and new references were added.
Comment 17: Section 3.3.2.: the authors should determine microsphere particle size distribution, since it is a critical parameter that influences efficacy of nasal drug delivery. It is not enough to have a mean value and SD since guidelines state that it is crucial that the majority of particles for nasal drug delivery is larger than 10 μm.
Response 17: Microsphere particle size distribution was discussed in the text as suggested by the reviewer.
Comment 18: Figure 8. Please check the legend in the figure, the figure currently does not match the discussion. Maybe the samples are switched.
Response 18: The figure was revised. As stated in the manuscript the incorporation of the nanoparticles (NP1-DFP) into the composite structure led to further delay in the release of DFP from the microspheres (M-NP1-DEP).
Comment 19: The section 3.3.6. is also missing references. Furthermore, as mentioned in the comment for the Methods sections, it is not completely clear how the detachment force was measured. Please elaborate and compare the obtained results with literature.
Response 19: The force required for detachment of the tablet from the lower surface after certain time of contact was measured as a function of displacement, by lifting the glass support at a constant rate of 1 mm min -1 until total separation of the components was achieved.
Comment 20: Lines 621 – 623: The authors state that the particle size of around 6 μm is appropriate for nasal administration, however, as mentioned earlier in the comments, the majority of particles need to be over 10 μm. The authors should correct this conclusion.
Response 20: The size of the particles determines whether they will be retained in the nasal cavity or inhaled into the lungs. It is stated in many literary sources that for pulmonary target particle size should be <5 μm, while particles larger than 5 μm are preferred for nasal administration. On the other hand smaller particle size (with increased surface area) leads to enhanced mucoadhesion ability, water uptake and dissolution rate, which may reflect in higher and faster absorption. Therefore, particle size must be optimized to attain desired mucoadhesion and dissolution properties while maintaining a deposition profile according to the required outcome. In the manuscript we have cited a study that proves that maximum olfactory deposition was observed with particles in the size range of 8 to 12 μm (https://doi.org/10.1089/jamp.2019.1549). Spray drying is a method that produces microparticles in a relatively wide range of sizes (which is also evident from the added particle size distribution), therefore due to the average size of the proposed microparticle models, which is very close to the indicated interval, we consider that they are suitable for nasal administration. However, we respect the recommendation of the reviewer and the conclusion was revised as suggested.
Round 2
Reviewer 2 Report
Comments and Suggestions for Authors/
Author Response
No reviewer's comments were found in Round 2.